# Where to flee? Preferences for host communities among displaced people in Congo

Nik Stoop[1], Peter Van der Windt[2]*, Sigrid Weber[3]

1 Institute of Development Policy, University of Antwerp, Research Foundation Flanders, Antwerp, Belgium, 2 Division of Social Sciences, New York University Abu Dhabi, Abu Dhabi, United Arab Emirates, 3 School of Politics, Economics and Global Affairs, IE University, Madrid, Spain

☯ These authors contributed equally to this work.
* petervanderwindt@nyu.edu

## Abstract

Previous research has explored host communities' attitudes toward displaced individuals, but much less is known about what displaced people seek in a host community. Understanding both perspectives is key to fostering successful local integration. We address this gap through a randomized conjoint experiment in which nearly 2,000 respondents in the Kasai region of the Democratic Republic of Congo - internally displaced persons (IDPs), returned IDPs, repatriated refugees, and members of the host community - evaluated hypothetical host communities, imagining where they would prefer to settle if forced to flee. Results indicate that, beyond safety, respondents prioritize job opportunities while also valuing social networks and political participation. Preferences are largely similar across groups. The findings suggest that the promotion of economic opportunities and inclusive governance are among the most effective strategies to create conditions conducive to local integration.

## Introduction

Forced displacement is a pressing global challenge. At the end of 2024, 123.2 million people were displaced due to conflicts or natural disasters [1]. Leading international and humanitarian organizations propose three durable "solutions" to the displacement challenge: resettlement in a third country, voluntary repatriation, and integration within host communities [2]. Resettlement, however, remains unattainable for most displaced individuals, and returning home is often impossible due to the persistent nature of conflict [1]. With global violence on the rise [3], local integration within host communities is becoming an increasingly important pathway for displaced individuals seeking opportunities to rebuild their lives.

Successful integration depends on aligning the needs of displaced individuals with the capacities and priorities of the host communities [4]. Existing research on displaced-host interactions largely emphasizes the preferences of host communities toward migrants (see, e.g., [5,6]). Considerably less attention has been paid to the preferences of those displaced. Given that displaced people actively make movement

**Data availability statement:** All data and instruments, including consent text, are available online (https://dataverse.harvard.edu/dataset.xhtml?persistentId=doi:10.7910/DVN/PHKNSU).

**Funding:** NS and PW received funding from the World Bank - UNHCR Joint Data Center. The funders did not play a role in the study design, data collection and analysis, decision to publish, or preparation of the manuscript.

**Competing interests:** The authors have declared that no competing interests exist.

decisions and exercise agency in choosing host communities [7,8], understanding their preferences is equally vital to promote integration. Recent studies have begun to examine the movement and destination preferences of displaced populations (e.g., [9–11]), while others have developed mechanisms to match refugees with host communities based on expected integration outcomes and settlement preferences (e.g., [12,13]).

We contribute to this emerging body of research by conducting a conjoint experiment in which we manipulate the characteristics of hypothetical host communities. This design allows us to assess the impact of host community characteristics on individuals' preferences for these host communities. Our research takes place in the Kasai provinces of the Democratic Republic of Congo (DRC), a country that is currently home to the world's fourth largest population of internally displaced people (IDPs) [14]. Between 2016 and 2017, the Kasai region experienced intense conflict, which displaced over 1.4 million people within the DRC and forced approximately 35,000 people abroad [15]. Although active fighting decreased after 2017, pockets of insecurity remain, occasionally triggering further displacement [16,17]. At the time of our data collection, in 2022-2023, hundreds of thousands of people remained internally displaced, and the region continued to struggle with high levels of poverty and food insecurity [18].

In collaboration with UNHCR, the UN Refugee Agency, we conducted a large-scale survey among people displaced by the conflict - IDPs, returned IDPs, and repatriated refugees - and members of the host community. As part of the survey, nearly 2,000 respondents completed a conjoint experiment to assess preferences for host communities beyond safety, focusing on job opportunities, church membership, family and language ties, and participation in local decision-making. Results indicate that respondents prioritize self-reliance, placing particular value on employment, while also seeking an active social and political life by participating in the church community and local political decision-making. Preferences are broadly consistent by respondents' displacement background, although IDPs and repatriated refugees show slightly stronger preferences for employment and social connections compared to host community members.

These findings have important policy implications. Although certain factors, such as family or language ties, are difficult to change, others can be actively leveraged to support the integration of displaced populations. We further explore the perceived integration mechanisms, and find that creating employment opportunities and stimulating participation in local decision making are among the most effective strategies. These factors not only enhance individuals' sense of economic contribution and agency, but also foster feelings of being welcomed, safe, and trusting toward the host community. By prioritizing these adaptable factors, policymakers and humanitarian organizations can create more conducive conditions for successful local integration.

## Preferences for host communities

Although preferences for host communities are understudied, the broader phenomena of displacement flows and destination choice have received considerable attention. Most prominent, conflict and violence have been found to shape when populations flee, when they return, and where they go (e.g. [19–21]). However, this body of literature also highlights the role of other factors in destination choice. We discuss these below and use them as the theoretical foundation to analyze individual preferences for host communities.

*Economic opportunities* are an important driver. Refugees tend to move to economically prosperous countries [22], likely due to their more generous welfare systems and the combination of low unemployment and high economic growth, which makes finding a job easier [23]. Similar patterns are observed among IDPs, who often move from remote or less developed areas to industrial centers and cities within the same country [24,25].

*Social factors* are also important, as they may reduce transition and adaptation costs. Refugee movements have been linked to colonial ties [26] and ethnic links [27] with the destination country. Additionally, the presence of existing migrant communities in their home country can increase the appeal of a destination [23]. At a more local level, displaced individuals also tend to move to areas where residents share their ethnic or political identity [28,29].

*Political factors* may further influence destination choice. Displaced people may seek a voice in local decision-making, while host communities may regulate how many refugees or IDPs they host and the extent of political integration they allow. Community-level programs in Lebanon, Jordan, and the DRC suggest that joint decision-making between displaced and host communities can promote social cohesion [4].

While previous research focuses mainly on aggregate displacement flows, this study directly measures respondents' preferences for host communities, across four population groups, using a randomized conjoint experiment in the Kasai region of the DRC. This approach provides novel, causal evidence on the relative importance of economic, social and political factors in host community choices. We pre-specified the following three hypotheses in our analysis plan (OSF registry: https://osf.io/7dp35):

H1: Households prefer host communities where they can secure a livelihood.
H2: Households prefer host communities where they have social, cultural or relational ties.
H3: Households prefer host communities where they can actively shape political decisions.

Households' displacement histories may shape preferences for host communities. Currently displaced individuals may prioritize immediate needs, such as economic opportunities, while households that were never displaced or returned long ago may place relatively more emphasis on social networks and political participation. We therefore also pre-specified to investigate the above hypotheses across internally displaced persons (IDPs), returned IDPs, repatriated refugees, and members of the host community.

## Context and research design

### Conflict and displacement in the Greater Kasai

We set out to test these hypotheses in the context of the Kasai, Kasai-Central, and Kasai-Oriental provinces in the Democratic Republic of Congo (panel (a) in Fig 1). This region was the epicenter of the 2016-2017 Kasai conflict, which displaced over 1.4 million people within the DRC and forced approximately 35,000 refugees to seek safety in Angola [15]. Most IDPs did not seek safety in camps or cities, but sought refuge with host families in rural settlements [30]. Today, large parts of the Kasai population remain displaced, and the region is characterized by food insecurity, ethnic tensions, and occasional instances of violence (e.g., [18,31]). A more detailed account of the history of the conflict can be found in S1 Appendix.

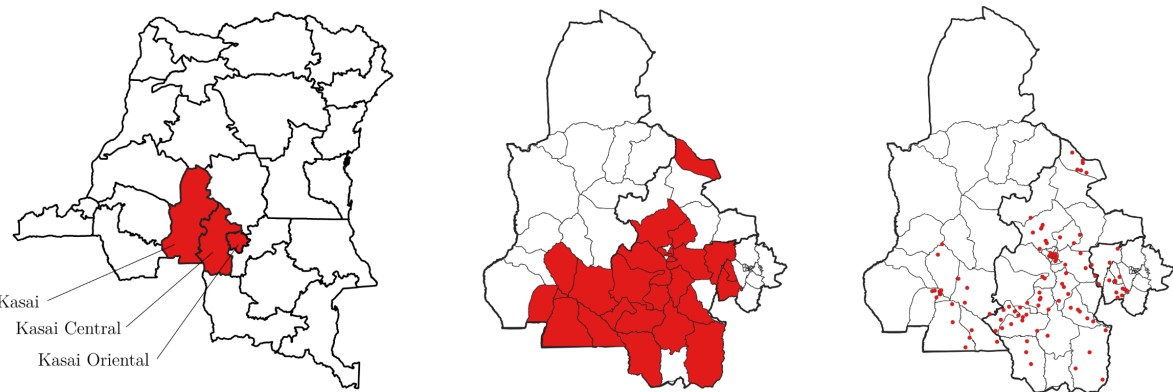

**Fig 1**. **Map of research area.** Notes: Panel (a): DR Congo and the three Kasai provinces where the study took place. Panel (b): The 62 health zones with the 27 selected health zones. Panel (c): The 119 survey locations. Maps created by the authors in QGIS using public domain shapefiles from the Humanitarian Data Exchange (HDX, OCHA).

## Data and sample

In collaboration with UNHCR, we conducted a survey representing both individuals displaced during the Kasai conflict - IDPs, returned IDPs and repatriated refugees - and members of host communities, defined as those with no history of displacement. We first selected the 27 health zones in the Kasai provinces most affected by displacement; i.e., those with particularly high numbers of IDPs and returned IDPs (panel (b) in Fig 1). Since there is no complete and up-to-date list of villages and towns in the Kasai region, we made use of Google AI satellite imagery that maps dwellings (these data are freely available online: https://sites.research.google/open-buildings). We defined 'localities' as geographical units of one square kilometer that contained at least 100 dwellings. Of the 5,302 identified localities, we randomly selected 119 study localities proportional to the number of dwellings in the health zones and localities. Next, we conducted a listing survey of all households in these selected localities (panel (c) in Fig 1) between July and November 2022. We randomly selected 4,069 from a total of 33,500 households. Approximately half of them (n = 1,965) were assigned to participate in the conjoint experiment that we analyze below. A detailed discussion of the sampling procedure can be found in S2 Appendix.

Survey data were collected between 24 November 2022 and 16 February 2023. The survey was commissioned by UNHCR to obtain insight into the socio-economic situation of the Kasai provinces. The data were collected by 53 experienced interviewers from the Congolese National Institute of Statistics (INS) who had followed an intensive two-week training. Voluntary and informed consent was obtained from each locality's chief, and subsequently from all survey participants. Consent was provided verbally due to high levels of illiteracy. All data and instruments, including consent text, are available online (https://dataverse.harvard.edu/dataset.xhtml?persistentId=doi:10.7910/DVN/PHKNSU). The study followed the Principles and Guidance for Human Subjects Research [32]. Ethics approval from the Official University of Bukavu in Congo (UOB/CEM/007/2022) and New York University Abu Dhabi (HRPP-2022-34).

The final sample for the conjoint experiment consists of IDPs (22%), returned IDPs (12%), repatriated refugees (39%), and members of the host community (26%). Half of the respondents are men (50%). The typical respondent is married (73%), has attended (88%) and graduated from primary school (74%), and can read (77%). One third of respondents (32%) do not speak Tshiluba, the primary local language in the study region, as their mother tongue. Approximately 50% of the respondents did not work in the week prior to the survey, and 85% report not having a regular job. A quarter of respondents (26%) attended a community meeting the preceding year. While 89% of respondents are Christian, there is

much variation among respondents; Revivalist (25%), Protestants (23%) and Catholic (15%) are the three largest denominations. In S3 Appendix, we provide definitions and summary information for all variables used in this study. That section also provides summary information by respondent group, including a test of differences; we find that, overall, the four groups are substantively very similar.

### Conjoint experiment

We used a conjoint experiment to measure individual preferences for host community characteristics. As part of the survey, respondents were presented with two potential host communities that randomly varied across five attributes: job availability (to measure the economic dimension of integration), church membership (social integration), family ties (relational integration), language ties (cultural integration), and the ability to participate in village decisions (political integration). To ensure that the hypothetical profiles are realistic, the attributes, corresponding levels, and drawings were carefully piloted and adjusted between June and October 2022 (the five months preceding the survey). Fig 2 provides an overview of the attributes and their levels, represented by drawings from a Congolese artist. To ensure comprehension, the survey was conducted in Tshiluba (although this language is not the mother tongue for a third of respondents, all respondents understand this language; e.g., the second-most common mother tongue in our sample is Bubindi, which is a dialect of Tshiluba). When presented with the two community profiles, respondents were asked to imagine having recently fled violence and to choose which of the two safe host communities they would prefer to live in. In this experiment we thus take it as given that the potential host communities are safe. Conditional on safety, displaced people have a choice among different hosting communities and face trade-offs in terms of economic, social, cultural or political integration. Participants completed two rounds of the conjoint experiment, involving a total of 3,930 forced choices (i.e., 7,860 observations). To avoid ordering effects, the order of attributes was randomized between respondents, but fixed across rounds for the same respondent.

### Inclusivity in global research

Additional information regarding the ethical, cultural, and scientific considerations specific to inclusivity in global research is included in the Supporting Information.

## Results

### Which host communities are preferred?

Fig 3 presents Average Marginal Component Effects (AMCEs), showing the causal impact of each attribute on a respondent's host community choices [33]. Dots represent the AMCE point estimates and thin (thick) horizontal lines show 95% (90%) confidence intervals. Replication material for all results are available online (https://doi.org/10.7910/DVN/HGIFXF).

Across all attributes, securing a livelihood is the most important consideration. The probability of selecting a host community increases by 10.5 percentage points (pp, $p < 0.01$) when work opportunities are available compared to when they are not. Social and cultural factors are also important: being able to join a local church and having family and language ties in the community significantly increase the attractiveness of host communities (by 6.8, 5.9 and 4.1 pp, $p < 0.01$, respectively). Finally, respondents value political participation in the host community: the ability to attend village meetings and participate in local decision-making increase the likelihood of choosing a host community by 5.2pp ($p < 0.01$).

A formal statistical test confirms that the availability of work opportunities is significantly more important than each of the other attributes (Table S4-4). Differences across the other attributes are not statistically distinguishable.

It is possible that these attributes reinforce or weaken each other. Fig S5-1 in the appendix shows that there are no significant interaction effects, with one exception: the ability to work and the ability to attend a local church appear to reinforce each other.

Job availability
(Economic integration)

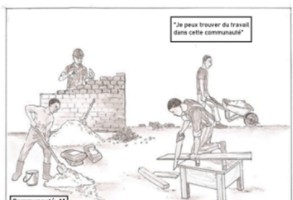 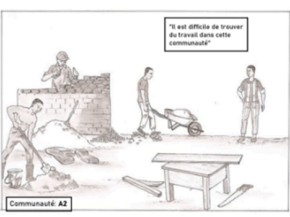

"You can get work in this village" "It is difficult to get work in this village"

Church membership
(Social integration)

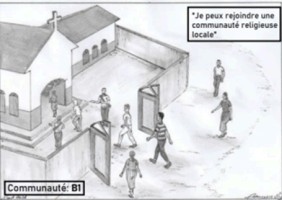 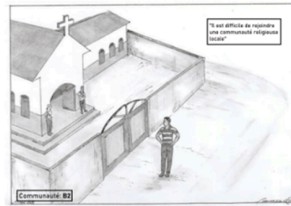

"You can join the local church" "You cannot join the local church"

Family ties
(Relational integration)

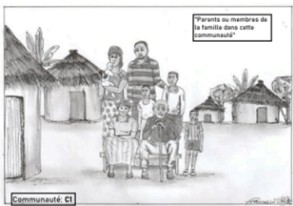 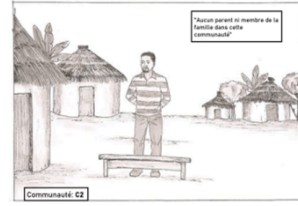

"You have relatives in the community" "You do not have relatives in the community"

Language ties
(Cultural integration)

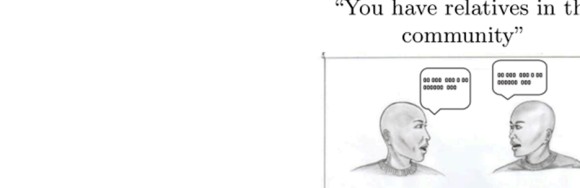 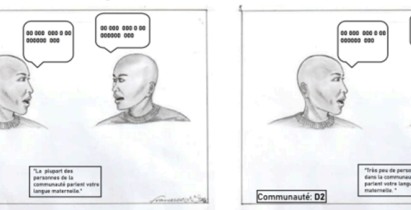 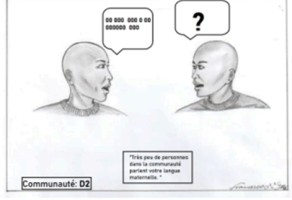

"The majority of the community speaks your mother tongue" "Few people in the community speak your mother tongue"

Participate in
village decisions
(Political integration)

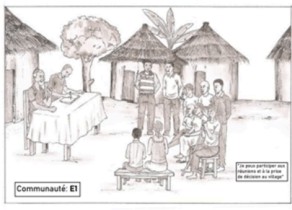 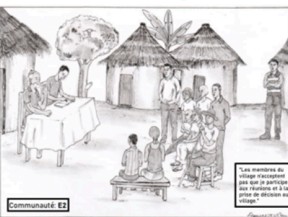

"You can attend village level meetings and participate in decision-making" "Village members do not want you to come to village meetings"

**Fig 2. Drawings conjoint experiment.** Notes: Drawings created by Francesco Nyamo; a Congolese artist. Reprinted under a CC BY license, with permission from Francesco Nyamo, original copyright 2022. The experiment's attributes, corresponding levels, and the drawings were carefully piloted and adjusted between June and October 2022 (the five months preceding the survey).

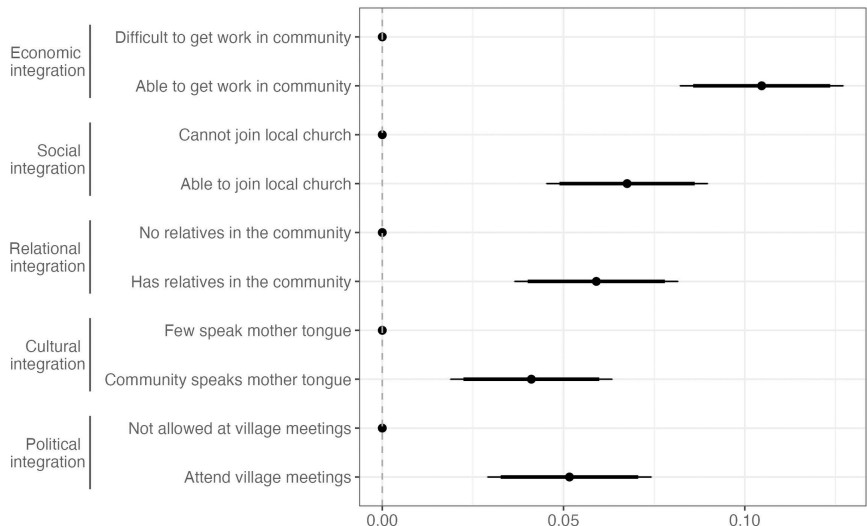

**Fig 3**. **Preferences for host community.** Notes: Average Marginal Component Effects for Host Community Choice (N = 7,860). Dots are the AMCE point estimates and thin (thick) horizontal lines show 95% (90%) confidence intervals. Full numeric results can be found in Table S4-1.

## Why are these host communities preferred?

To understand why these host community attributes shape preferences, the survey included follow-up questions designed to capture potential mechanisms. After selecting their preferred community, respondents evaluated five statements for each community presented: i) I could contribute to the economy of this community, ii) I would feel welcomed in this community, iii) I would feel safe in this community, iv) I could trust the members of this community, and v) I could improve this community by bringing in new ideas and cultures. Responses were measured on a five-point Likert scale from "not at all" to "strongly agree".

Fig 4 shows that respondents perceive economic and political integration as distinct and important dimensions of life in a host community. Communities offering job opportunities increase respondents' perceived ability to contribute to the economy by 0.8 points on the five-point scale (p < 0.01). Similarly, the ability to participate in local decision-making increases the perceived likelihood of contributing ideas to the community by 0.7 points (p < 0.01). These findings suggest that economic and political participation operate largely independent rather than reinforcing each other.

Both economic and political attributes also enhance respondents' perceptions of safety, feeling welcome, and trust toward the community. Social attributes such as joining a local church or having relatives in the community have moderate effects, whereas speaking the same language has the least impact on these perceived integration mechanisms. This aligns with the conjoint experiment results, where language ties also has the lowest AMCE point estimate. Overall, while work opportunities remain the strongest driver of host community preferences, political, social, and cultural factors also contribute to respondents' sense of belonging and agency, helping to explain why they are also valued in the conjoint experiment.

## Preferences by displacement experience

By design, our sample included individuals displaced during the Kasai conflict - IDPs, returned IDPs, and repatriated refugees - as well as members of the host community (Fig S2-1 in the appendix explains how these groups were defined). Fig 5 examines whether displacement history shapes preferences for host communities, using AMCEs. We present

**Fig 4.** **Effect of community attributes on perceived integration mechanisms.** Notes: Average marginal component effects for integration statements (N = 7,860). Dots are the AMCE point estimates and thin (thick) horizontal lines show 95% (90%) confidence intervals. Full numeric results can be found in Table S4-2.

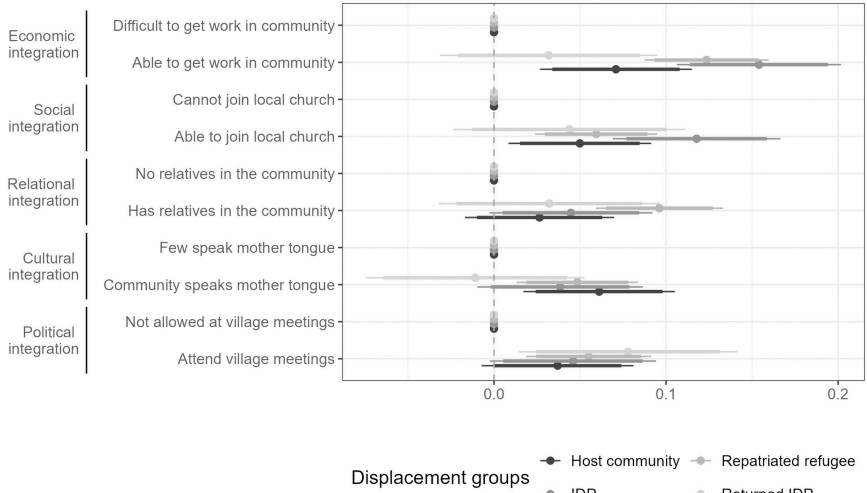

**Fig 5.** **Preferences for host community by displacement experience.** Notes: Average Marginal Component Effects for Host Community Choice (N = 7,860) by displacement experience. Dots are the AMCE point estimates and thin (thick) horizontal lines show 95% (90%) confidence intervals. Full numeric results can be found in Table S4-3.

marginal means for the core results in the appendix given the difficulties in interpreting and aggregating preferences for different subgroups when using AMCEs (e.g., [34,35]).

Overall, preferences are broadly similar across groups. Formal statistical tests do not show significant differences across the three populations with a displacement history. However, compared to host community members, IDPs express a stronger preference for work opportunities, consistent with more immediate livelihood needs. In addition, IDPs show a stronger preference for being able to join a local church, while repatriated refugees place greater value on communities

with relatives (Table S4-5). Returned IDPs display the weakest effects on most attributes. This may partly reflect substantive differences, but it could also be due to their smaller group size, which produces larger standard errors relative to the other groups.

Why are preferences otherwise so similar across groups? One explanation is that households in the Kasai provinces face harsh living conditions regardless of displacement history. Indeed, displaced and non-displaced households in our sample are very similar across key indicators such as employment, food security, family structure and health (see Table S3-2). Another explanation relates to the design of the conjoint experiment: respondents were asked to imagine having recently fled violence. Members of the host community - who also faced considerable violence during the Kasai conflict - may be equally able to put themselves in the position of those who experienced displacement.

## Robustness

In S6 Appendix we undertake several robustness tests. We obtain similar results when estimating marginal means [34] instead of AMCEs, and when using a continuous outcome measure in which respondents rank how much they would like to live in each presented host community (rather than the forced choice outcome). Our findings are also robust to controlling for a large set of covariates. Finally, we find similar results across subgroups defined by mother tongue and respondent's literacy status.

## Conclusion

Integration within host communities is an increasingly important pathway for displaced individuals to rebuild their lives [2]. Although much research has investigated the preferences of host communities toward migrants, much less attention has been paid to the characteristics that displaced individuals themselves consider important in a host community. To address this gap, we partnered with UNHCR, the UN Refugee Agency, and conducted a conjoint experiment embedded in a large-scale survey in the Kasai provinces of the Democratic Republic of Congo. Nearly 2,000 respondents - including IDPs, returned IDPs, repatriated refugees, and members of the host community - evaluated hypothetical host communities, imagining where they would prefer to settle if forced to flee.

Our findings indicate that while respondents prioritize securing a livelihood, they also value building a social and political life. As such, they significantly prefer host communities that facilitate social integration - through church membership, family ties, and shared language - and that allow participation in local decision-making. These results suggest that fostering employment opportunities and encouraging political participation are key strategies for policymakers and humanitarian organizations to support local integration of displaced individuals. Such opportunities enhance individuals' sense of contribution and agency, while also promoting feelings of being welcomed, safe, and trusting toward the host community. We find that preferences among respondents with different displacement histories are very similar, suggesting that these findings may be broadly relevant to populations facing hardship.

We consider this study a first step, and encourage future research in two directions: i) investigating the role of other host community attributes - such as NGO presence, access to land, and availability of schools - across different contexts; and ii) better understanding the constraints that displaced populations operate under, which may lead to divergence between location preferences and actual location choices. While in developed countries displaced individuals often have little agency in host community selection (e.g., refugees may be assigned to host locations by government agencies [13]), in developing contexts choices are shaped by informational, financial, and logistical constraints. Measuring both preferences and actual host community characteristics, and examining why they differ is crucial for advancing our understanding of integration dynamics.

Given the already large scale of forced displacement - and the expectation of further escalation due to ongoing violence, desertification, rising sea levels, and more frequent severe weather events - it is essential to understand the

determinants of successful integration. The preferences of displaced populations for host community characteristics are one critical piece of this broader puzzle.

## Supporting information

**S1 Text. Supporting information for Where to flee? Preferences for host communities among displaced people in Congo. Filled Out Questionnaire on Inclusivity in Global Research.**
(PDF)

## Acknowledgments

We thank Mateo Villamizar Chaparro, Jens Hainmueller, J. Andrew Harris, Oguzhan Türkoglu, Jeremy Weinstein, Mark Marvin Kadigo, and participant at Stanford's Immigration Policy Lab, King's QPE workshop, and at the Violence, Instability, and Peace Workshop for comments. Thanks to Haoyu Zhai for research assistance. Thanks also to Victor Daye, Marguerite Duponchel, Mamadou Fofana, Emmanuel Kandate, Hyacine Manawa, Harriet Kasidi Mugera, Papa Moussa Ndoye, Sergiy Radyakin, and Michael Wild. We thank Congo's National Institute of Statistics and the surveyors for data collection. We would like to thank Francois Nyamo for the artwork used in this study.

## Author contributions

**Conceptualization:** Peter Van der Windt, Nik Stoop, Sigrid Weber.

**Formal analysis:** Peter Van der Windt, Nik Stoop, Sigrid Weber.

**Writing – original draft:** Peter Van der Windt, Nik Stoop, Sigrid Weber.

**Writing – review & editing:** Peter Van der Windt, Nik Stoop, Sigrid Weber.

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
