## [Decision Letter · Decision Letter 0]

6 Aug 2025

PONE-D-25-26296Where to flee? Preferences for host communities among displaced people in CongoPLOS ONE

Dear Dr. van der Windt,

Thank you for submitting your manuscript to PLOS ONE. After careful consideration, we feel that it has merit but does not fully meet PLOS ONE’s publication criteria as it currently stands. Therefore, we invite you to submit a revised version of the manuscript that addresses the points raised during the review process. The reviewers all provide comments which should be possible to address. I am happy for you to ignore the comments from reviewer 1 if you choose. 

We look forward to receiving your revised manuscript.

Kind regards,

Alison Parker

Academic Editor

PLOS ONE

Journal Requirements:

3. In the ethics statement in the Methods, you have specified that verbal consent was obtained. Please provide additional details regarding how this consent was documented and witnessed, and state whether this was approved by the IRB.

4. Please update your submission to use the PLOS LaTeX template. The template and more information on our requirements for LaTeX submissions can be found at http://journals.plos.org/plosone/s/latex.

5. Please expand the acronym “UNHCR” (as indicated in your financial disclosure) so that it states the name of your funders in full.

[We thank Mateo Villamizar Chaparro, Jens Hainmueller, J. Andrew Harris, Oguzhan Tüurkoglu,237 Jeremy Weinstein, Mark Marvin Kadigo, and participant at Stanford’s Immigration Policy 238 Lab and at the Violence, Instability, and Peace Workshop for comments. Thanks also to 239 Victor Daye, Marguerite Duponchel, Mamadou Fofana, Emmanuel Kandate, Hyacine Man- 240 awa, Harriet Kasidi Mugera, Papa Moussa Ndoye, Sergiy Radyakin, and Michael Wild. We 241 thank Congo’s National Institute of Statistics and the surveyors for data collection. We 242 thank the World Bank-UNHCR Joint Data Center on Forced Displacement and New York 243 Abu Dhabi for funding. We would like to thank Francois Nyamo for the artwork used in this 244 study. Ethics approval from the Official University of Bukavu (UOB/CEM/007/2022) and 245 New York University Abu Dhabi (HRPP-2022-34).]

[NS and PW received funding from the World Bank UNHCR Joint Data Center. The funders did not play a role in the study design, data collection and analysis, decision to publish, or preparation of the manuscript]

7. When completing the data availability statement of the submission form, you indicated that you will make your data available on acceptance. We strongly recommend all authors decide on a data sharing plan before acceptance, as the process can be lengthy and hold up publication timelines. Please note that, though access restrictions are acceptable now, your entire data will need to be made freely accessible if your manuscript is accepted for publication. This policy applies to all data except where public deposition would breach compliance with the protocol approved by your research ethics board. If you are unable to adhere to our open data policy, please kindly revise your statement to explain your reasoning and we will seek the editor's input on an exemption. Please be assured that, once you have provided your new statement, the assessment of your exemption will not hold up the peer review process.

8. We note that you have referenced (Marco Avi˜na, Taeku Lee, Mashail Malik, Reed Rasband, Marcel Roman, and Priyanka Sethy. Which immigrants do citizens prefer? a meta-reanalysis of 96 conjoint experiments. Unpublished working paper, 2024) which has currently not yet been accepted for publication. Please remove this from your References and amend this to state in the body of your manuscript: (Marco Avi˜na, Taeku Lee, Mashail Malik, Reed Rasband, Marcel Roman, and Priyanka Sethy. Which immigrants do citizens prefer? a meta-reanalysis of 96 conjoint experiments. Unpublished working paper, 2024) as detailed online in our guide for authors

9. Your ethics statement should only appear in the Methods section of your manuscript. If your ethics statement is written in any section besides the Methods, please delete it from any other section.

10. We note that Figures 1a, 1b, and 1c in your submission contain map images which may be copyrighted. All PLOS content is published under the Creative Commons Attribution License (CC BY 4.0), which means that the manuscript, images, and Supporting Information files will be freely available online, and any third party is permitted to access, download, copy, distribute, and use these materials in any way, even commercially, with proper attribution. For these reasons, we cannot publish previously copyrighted maps or satellite images created using proprietary data, such as Google software (Google Maps, Street View, and Earth). For more information, see our copyright guidelines: http://journals.plos.org/plosone/s/licenses-and-copyright.

1. You may seek permission from the original copyright holder of Figures 1a, 1b, and 1c to publish the content specifically under the CC BY 4.0 license. 

11. We note that Table 1 in your submission contains copyrighted images. All PLOS content is published under the Creative Commons Attribution License (CC BY 4.0), which means that the manuscript, images, and Supporting Information files will be freely available online, and any third party is permitted to access, download, copy, distribute, and use these materials in any way, even commercially, with proper attribution. For more information, see our copyright guidelines: http://journals.plos.org/plosone/s/licenses-and-copyright.

1. You may seek permission from the original copyright holder of Table 1 to publish the content specifically under the CC BY 4.0 license.

In the table caption of the copyrighted figure, please include the following text: “Reprinted from [ref] under a CC BY license, with permission from [name of publisher], original copyright [original copyright year].”

12. Please include captions for your Supporting Information files at the end of your manuscript, and update any in-text citations to match accordingly. Please see our Supporting Information guidelines for more information: http://journals.plos.org/plosone/s/supporting-information.

Reviewers' comments:

Reviewer's Responses to Questions

**Comments to the Author**

1. Is the manuscript technically sound, and do the data support the conclusions?

Reviewer #1: Yes

Reviewer #2: Partly

Reviewer #3: Yes

Reviewer #4: No

2. Has the statistical analysis been performed appropriately and rigorously?

Reviewer #1: Yes

Reviewer #2: Yes

Reviewer #3: Yes

Reviewer #4: Yes

3. Have the authors made all data underlying the findings in their manuscript fully available?

Reviewer #1: Yes

Reviewer #2: Yes

Reviewer #3: Yes

Reviewer #4: Yes

4. Is the manuscript presented in an intelligible fashion and written in standard English?

Reviewer #1: Yes

Reviewer #2: Yes

Reviewer #3: Yes

Reviewer #4: Yes

5. Review Comments to the Author

Reviewer #1: The authors should considere this two papaers:

1. Family and Religion in Luba Life:

Centrality, Pervasiveness, Change and Continuity

Tshilemalema Mukenge (AUTHOR)

2. Pouvoir, territorialité et conflictualité au Grand Kasaï (République démocratique du Congo)

Power, territoriality and conflicts in Large Kasaï (Democratic Republic of Congo)

Kabata Kabamba (AUTHOR)

The two paper wil help to reinforce the similarities explanations between IDPs and host communities, and will shed some light on the IDPs preferences as well.

Reviewer #2: Where to flee? Preferences for host communities among displaced people in Congo

This study is an interesting report on factors driving the migratory preferences of Congolese. It contributes important insights to policymakers and practitioners working on migration.

Overall, this work should be published in PlosOne. I think the MS could benefit from some reframing and added clarity in the text because there are meaningful differences among subpopulations (displaced versus non), as well as fixing of minor errors. I put some suggestions for additional analyses that should either be added to the main text or put into the appendix, depending on what they result in.

Framing/Contribution

I have some reservations about how the study is presented. It feels somewhat like the authors were struggling to cut words, and ended up cutting too much. Perhaps this also led them to frame the paper in what may have been the most parsimonious presentation, but to me, the current framing doesn’t do justice to the study sample nor its findings.

I was confused for quite a while as to what types of people this study is about. The introductory framing leads the reader to believe that the paper will be about refugees and/or displaced persons seeking international integration. After reading the manuscript I thought it was about IDPs, but understood it also had some non-displaced persons mixed in. Yet, from the SI materials I found out that the sample actually comprises refugees, IDPs, and those who remained despite conflict. This really needs to be made clear in the main text of the MS and I think it needs to be more clearly addressed even in the abstract and introduction. How do these three groups differ from one another and how are they the same, especially with regards to the experimental outcomes?

I note this because the framing lumps IDPs with refugees and I question if this is a reasonable generalization to make. IDPs are a special group differentiated first, by the fact that they remain in their own country where they share a language (or usually at least some linguistic similarities) and cultural ties to a higher degree at least than they would had they crossed an international border (or at least this is the most likely case); and second, by the fact that they may be those who couldn’t afford to go abroad - so they may be poorer and less educated than refugees. For instance, most refugees who came to Europe during the 2015 ”crisis” are single males from wealthier backgrounds compared to those who remained in the region or were internally displaced. It seems also like those who fled abroad might have been more likely to actually experience some of these situations firsthand and especially things like that the community does not speak one’s mother tongue.

The authors run an analysis of displaced versus not, what about returned refugees versus IDPs versus those who remained? I would like to see the results for actual refugees compared to not displaced as well to confirm all the claims that they are not different are confirmed.

We also already see in the SI file that displaced persons (lumping refugees and IDPs together) are indeed diff from those who were never displaced. The latter are not significantly affected by being allowed at community meetings nor by relatives in the area. The MS says these differences are not significant but I couldn’t find an actual test to verify that. Regardless if the two groups are not significantly different from each another, (this could be due to lack of power since the non-displaced sample is small), the MS should be more honest about what it’s studying and what it finds. These subgroup differences should not simply be ignored.

This is important mainly because one could question how much fleeing from violence affects the outcomes at all and whether some of the preferences are just preferences about migration in general? The sample is largely suffering from a lack of regular income, so available jobs seems to be a desirable feature for all, regardless of displacement or not. The authors might want to reframe around these distinctions. For the sub sample of non-displaced, anything that is sig seems to matter for non displaced people as well as displaced. For displaced persons, other factors like family and attending village meetings are important when they are not for non-displaced persons. What might this indicate? Perhaps looking at the mechanisms by subgroups of IDPs, refugees, and those who remained would be helpful?

An additional robustness check for the experimental analysis would be to show results with controls for all potential differences, including one for IDPs, refugee, and never displaced but also the other indicators of household economics, gender, etc. Demonstrate how these potential differences don’t matter for the experiment. Table S5 doesn’t really accomplish this. But also since the authors have this information already separated out, the reporting of sample characteristics in the MS should be broken down by those who fled and those who did not, or even more preferably, IDPs, refugees, and never displaced.

Experimental design:

The design could use some more explanation of how realistic these hypothetical safe havens to flee to are. How often do villages hold meetings and collect preferences of villagers? And how common is it that they would bar residents (or even that they would allow temporary residents to attend??) Do all respondents actually have family they could move to? Might not all their family live in the same village? Would a church really deny other Christians from participating? What kind of church is that?Along these lines: Are all respondents Christian? There’s nothing on religion or how often people attend church. That seems bizarre given that church attendance is an experimental arm.

The term ”coethnic” is not well defined nor theorized. All three attributes of co-linguists, relatives, and co-religion can be considered to be coethnics. Why are there no indicators for language or religion in the samples? Particularly if language group is how ethnicity is defined in Congo - why are there no analyses controlling for ethnic group or language? Given that the experiment manipulates language, is it likely for some language groups that no one in a village would speak their language? Some African languages are very close to others, so this may not be a big problem. Minority language native speakers are also probably more likely to speak another more popular language. This begs the question: what is shared language doing in the experiment and how does this vary across sub populations?

The pictures accompanying the experiment don’t just depict the points about lack of jobs or family or ability to get into meetings and churches, but also loneliness. They’re a bit misleading from the main trigger/mechanism because they emphasize a person being left all alone. Like even if there aren’t jobs, one could still volunteer help. Even if there’s no family, a person might be able to make friends. One robustness check here would be to see if the illiterate (who maybe relied more on the pictures) differ from those who are literate. But, for instance, the arms on attending church and village meetings are also just about exclusion. For meetings - is it about not being welcome or about not being able to join in decisions? The authors attempt to address some of this in the section on mechanisms, but the logic behind this section is not well explained. A clearer explanation of these analyses and the motivation for them is needed. Be clear how this section addresses weaknesses in the design.

But I also worry that since the three experimental manipulations of church, family, and shared language all get at social integration whereas there is only one arm for economic and political integration, so that claiming these other two mattered more could be problematic. Couldn’t it be that each social integration arm is overlapping with the other two in its effects? So some of the effect is being soaked up by the other aspect(s)? Comparing which matters more - that your relatives are there or the community speaks your language could contradict somewhat. Your relatives may be expected to speak your language. So if the respondent gets the arm with few speak their language but their relatives are there, this detracts/weakens the effects of the shared language arm. Can the authors convincingly address this? Otherwise they may want to do away with the claim that economic and political integration matter more than social.

Perhaps this final point could be addressed by checking ACIEs between the experimental arms? Are any of the arms enhanced by the others? It seems shared language will matter community meetings and church, for example.

Are figures 3 abd 4 diff? 4 should show us something on the displaced versus not but I think it’s the same figure.

The authors should also make the pre-analysis plan available. This manuscript is not anonymized so there’s no reason it shouldn’t be provided, but another approach could be an anonymized version within the SI materials.

Smaller issues:

- Page 4 has at least one typo calling the experiment a ”joint” experiment. It also could use a look in terms of writing flow. Some of the points made there seem out of place - like voting for right wing parties and asylum applications. The link between that setting and the Congo seems rather tenuous.

- Page 9 I think 90% is missing from the sentence for the thin lines …

- Food insecurity measure seems to be off - it should be a number between 0 and 7 as I understand the question but the max is reported as 35 and the mean is 11?

- Sampling - what does most affected by displacement mean? Places where people fled or fled to or both?

Reviewer #3: The main issue of the article is the forced correlation between refugee and IDPs. This leads to confusion and miss-led comparison about refugee studies and reasons to move, and the myriads of reasons sustaining their decision making (to flee, and to where) are not like those of displaced people. To review based the Kampala Convention and 951 Refugee Convention and its 1967 Protocol 951 Refugee Convention and its 1967 Protocol. With this, minor suggestions and changes are needed.

Reviewer #4: What characteristics do displaced people value in looking for a place to settle? The authors use a conjoint survey experiment to ask about 2000 respondents living in a region of the Democratic Republic of the Congo that has been impacted by internal forced displacement. 74% of these respondents are internally displaced people (IDPs). In the conjoint, they examine the following 5 attributes: whether work is available or not, a local church or not, family members or not, local language or not, and ability to participate in village meetings or not. This is an impressive data collection effort, and I am curious about the other insights that this survey might provide.

With respect to the main conjoint experiment, I suggest that the authors need to be more precise about the interpretation of their findings and do formal hypothesis testing to verify their claims. Although the authors claim that they find work opportunities is most important, more than coethnicity or family, the actual results in Figure 2 show that all 5 attributes may be equally important — the AMCE for work seems slightly more important than the others but that is not formally tested.

The conjoint has a very simple setup. Within each of the 5 attributes, there are only two levels and one level is always obviously preferred to the other — i.e. opportunity to work is better than no work, local church is better than no church, having relatives is better than not, having a common language is better than not, being able to attend village meetings is better than not — holding all the other attributes constant. The marginal means shown in Fig. S3 and the conjoint results with a continuous preference measure in Fig. S4 seem to the show the same as the main result.

Thus, the important comparisons are not within each attribute, but rather across attributes — comparing the importance of work against church against relatives, etc. But from Figure 2, the CIs seem to overlap, perhaps work stands out a little, but that will need to be formally shown. Are the differences between the estimates statistically significant?

It may be useful for the authors to reference this 2022 AJPS article: “What Do We Learn about Voter Preferences from Conjoint Experiments?” by Abramson, Kocak, and Magazinnik to how to interpret the AMCE precisely.

Originally, when I started reading this paper, I had generalizability concerns — can the authors discuss how these findings generalize to other displacement contexts and from IDPs to refugees, for example? But seeing these results, I wonder if they would extend to many (non-displaced) populations, not just a vulnerable one experiencing insecurity. Indeed, the authors find similar results comparing the IDPs with the host community members in their sample.

I also wonder why the authors chose to focus on these 5 attributes. There are others that this literature discusses notably, availability of land, access to markets, and access to humanitarian aid, for example. Of course any conjoint, especially if the authors needed to keep it simple, need to prioritize some attributes over others, but I wonder if adding either of these other important attributes would shift the results so that some attributes would stand out — for example a stronger preference for work rather than aid.

Finally, I think the authors can expand Section 2: Preferences for Host Communities more so that it does more than simply list and summarize existing studies. This section can highlight the unique contributions of this paper as well as list the hypotheses.

6. PLOS authors have the option to publish the peer review history of their article (what does this mean?). If published, this will include your full peer review and any attached files.

Reviewer #1: **Yes: **GLORIA NGUYA BINDA

Reviewer #2: No

Reviewer #3: No

Reviewer #4: No

---

## [Author Response · Author response to Decision Letter 1]

10 Oct 2025

Please find our detailed response letter attached

---

## [Decision Letter · Decision Letter 1]

11 Nov 2025

Where to flee? Preferences for host communities among displaced people in Congo

PONE-D-25-26296R1

Dear Dr. van der Windt,

We’re pleased to inform you that your manuscript has been judged scientifically suitable for publication and will be formally accepted for publication once it meets all outstanding technical requirements.

Kind regards,

Alison Parker

Academic Editor

PLOS ONE

Additional Editor Comments (optional):

Reviewers' comments:

Reviewer's Responses to Questions

**Comments to the Author**

1. If the authors have adequately addressed your comments raised in a previous round of review and you feel that this manuscript is now acceptable for publication, you may indicate that here to bypass the “Comments to the Author” section, enter your conflict of interest statement in the “Confidential to Editor” section, and submit your "Accept" recommendation.

Reviewer #1: All comments have been addressed

Reviewer #3: All comments have been addressed

2. Is the manuscript technically sound, and do the data support the conclusions?

Reviewer #1: Yes

Reviewer #3: Yes

3. Has the statistical analysis been performed appropriately and rigorously?

Reviewer #1: Yes

Reviewer #3: Yes

4. Have the authors made all data underlying the findings in their manuscript fully available?

Reviewer #1: Yes

Reviewer #3: Yes

5. Is the manuscript presented in an intelligible fashion and written in standard English?

Reviewer #1: Yes

Reviewer #3: Yes

6. Review Comments to the Author

Reviewer #1: NO COMMENTS AS THE AUTHOR HAS CONSIDERED MY REVIEWS AND ALSO PROVIDED A BALANCED EXPLANNATION OF RESPONDENTS

Reviewer #3: No comments. All the comments from from the first round were addressed.

I also noticed the the authors will have to discuss in the future the difference between refugees and IDP and clarify the context.

Regards

7. PLOS authors have the option to publish the peer review history of their article (what does this mean?). If published, this will include your full peer review and any attached files.

Reviewer #1: No

Reviewer #3: No

---

## [Editor Report · Acceptance letter]

PONE-D-25-26296R1

PLOS ONE

Dear Dr. van der Windt,

I'm pleased to inform you that your manuscript has been deemed suitable for publication in PLOS ONE. Congratulations! Your manuscript is now being handed over to our production team.

Kind regards,

on behalf of

Dr. Alison Parker

Academic Editor

PLOS ONE